

# Technical Note: Space-Time Statistical Quality Control of Extreme Precipitation Observations

Abbas El Hachem[1], Jochen Seidel[1], Florian Imbery[2], Thomas Junghänel[2], and András Bárdossy[1]

[1]Institute for Modelling Hydraulic and Environmental Systems, University of Stuttgart, D-70569 Stuttgart, Germany
[2]Deutscher Wetterdienst, Offenbach, Germany

**Correspondence:** Abbas El Hachem (abbas.el-hachem@iws.uni-stuttgart.de)

**Abstract.** Precipitation extremes form the basis of many engineering design decisions. Extremes are rare events which may differ strongly from "normal" observations. Unfortunately some of the observed extremes may be inaccurate or false. The purpose of this investigation is to present a quality check of observed extremes using space-time statistical methods. As a first step the biggest values for each observation location and event duration are selected. For each of these the observed values of all other stations corresponding to the same time steps are collected and transformed using a Box-Cox transformation factor derived from a fitted truncated normal distribution. The value at the extreme location is estimated using the surrounding stations and the calculated spatial variogram, and this estimated value is compared to the observed extreme. If the difference exceeds the critical value of the test, the extreme is flagged as possible outlier. The same procedure is repeated for different aggregations in order to avoid singularities caused by convection. The flagged extremes are then compared to the extremes of the surrounding stations using the same procedure – interpolation and subsequent comparison of the interpolated and the observed values. Flagged extremes are subsequently compared to the corresponding radar and discharge observations and finally implausible extremes are removed. The procedure is demonstrated using observations of sub-daily and daily temporal resolution in Germany.

## 1 Introduction

A clear definition of an outlier might be intuitive to many but it has been formulated differently by several researches. In the work of Barnett and Lewis (1984) an outlier was defined as an observation showing an inconsistent behavior compared to other data values. Hawkins (1980) described an outlier as being an observation that differs substantially from other observations as if it might have been produced by an alternating mechanism. More precisely, for Iglewicz and Hoaglin (1993) an outlier is an observation that arouses suspicion to the analyst and does not belong to the same data distribution. Johnson et al. (2017) defined two types of outliers, namely those associated with an error and those associated with a real observation. The reasons for an observation being erroneous could either be due to instrumental errors (e.g. use of false instrument, equipment malfunction, false equipment operation) or/and human errors (false reading or recording or even computation of observations).

Hydroclimatological data are of unique nature as they occur in a non repetitive manner. If an observation is not registered correctly reconstructing such a measurement is very challenging, especially for extreme precipitation values. Due to the high





spatial and temporal variability of such events, surrounding rain gauges can often not be used for outlier detection or data
plausibility checks. However, observation of precipitation extremes are essential for flood analysis, extreme value statistics,
stationarity analysis and many other engineering practices. The presence of outliers in a data set can lead to under- or overesti-
mation of design values.

Precipitation observations have a space-time dimension. Observations are taken at different locations in space and in discrete
time intervals. Some precipitation events occur over a local spatial scale with short duration and high intensity. These events
are outliers defined as 'single' events. They were correctly observed but strongly differ from surrounding observations. These
observations play, for example, a major role in urban hydrology.

Even in normal conditions false observations can occur especially when considering high-resolution temporal data (1 minute
frequency). Many quality control algorithms have been developed and are being used by weather service agencies to minimize
and detect false measurements. Durre et al. (2010) established a comprehensive QC algorithm for daily surface meteorological
observations (temperature, precipitation, snow fall and snow depth). For precipitation data, the QC method for detecting false
observations consisted several steps from which a climatological outlier checks for flagging values exceeding a certain temper-
ature dependent threshold and a spatial consistency checks by which the target observation is compared to neighboring ones
and is flagged if the difference exceeds a certain climatological percent ranks threshold. Qi et al. (2016) implemented a QC al-
gorithm to identify erroneous hourly rain gauge observations by using additional information as radar quantitative precipitation
estimates (QPE). A common approach for detecting outliers is to use an interpolation method to estimate the suspected ob-
servation using the surrounding locations. When only comparing the observed to the estimated value, many observation might
not be correctly captured depending on the event spatial extent. In the work done by Hubbard et al. (2005) a QC method was
developed for daily temperature and precipitation consisting of four steps, observations are flagged if they do not fall within
+-3 standard deviation of the long-term mean and if they differ from the estimated value using spatial regression technique.
Some other methods are available but are often limited to time series analysis and tend to disregard the temporal-spatial extent
of precipitation. Moreover, due to the presence of non-negative and many no-precipitation (zero) values, precipitation data have
a positively non-normal skewed distribution with heavy tails (Klemeš, 2000) and fall under the zero-inflated data. An adequate
transformation of the data should be performed to reduce the effect of the data skewness. A relatively simple approach to nor-
malize a variable is to apply a Box-Cox transformation (Box and Cox, 1964). To cope with the positive nature of precipitation
a transformation to a truncated normal distribution can be used.

The following work proposes a statistical space-time methodology based on interpolation in a cross-validation mode to find
possible outliers in the precipitation data observations across several temporal aggregations. An outlier is defined here as an
observation that strongly differs for a certain temporal aggregation from it's spatial neighouring locations. A difficult task
while working with outliers in general and especially in hydrology is distinguishing between correct and false observations.
Therefore, to validate detected outliers, the suspected values are additional compared to external information such as discharge
and radar measurements is done.





This paper is organized as follows: after the introduction, the data and methodology to find possible outliers in the data are presented. Afterwards the results of the quality control procedure are presented and two examples of verification via subsequent comparison to radar or discharge data are shown. The paper ends with a discussion and a conclusion.

## 2 Study Area and Data

This Study was done using the German wide precipitation data set from the German Weather Service DWD which covers an area of approximately 357 000 $km^2$. The average annual rainfall in Germany is around 800 mm and can reach up to 2100 mm in the higher elevations of the Alps in the South. Currently, the DWD operates a network of rain gauges with different temporal resolutions ranging from minutely to daily (Deutscher Wetterdienst, 2020). Hourly and sub-hourly data are available from the 1990s onwards. The number of these stations has been continuously increasing since then. As for the daily data, some stations are available since the year 1900. After the year 1990, the number of stations with daily observations started to decrease while they were replaced by automated rain gauges. Rain gauges near the border without neighbours from other countries were not included in this analysis. A 30 kilometer inland buffer was used to select all rain gauges that are located within this region. This is relevant for the used methodology relying on neighbouring observations.

Figure 2 shows the change of the number of stations in Germany and the corresponding distribution of available observation years for daily and sub-hourly station data. In scope of this study, data have been collected for most available temporal resolutions (1 minute, 1 hour and 1 day) and time periods. In the 1990s, most DWD rain gauges were tipping buckets or drop counters. From 2000 onwards, these were replaced by weighing gauges (OTT Pluvio) and since 2017 these are being replaced by combined tipping bucket and weighing rain gauges (Lambrecht rain[e]).

Precipitation data from the recent DWD observation network go through several quality control steps. The first step is a quality control directly at the automatic stations. Since this is an automatic test, relatively wide thresholds are applied. It includes tests for completeness, thresholds, temporal and internal consistency. Based on these tests, a quality flag is assigned to the data. The data is then submitted to a database. Another test with tighter thresholds is then performed, based on the QualiMet software (Spengler, 2002). This phase of the quality check also tests for completeness as well as climatological, temporal, spatial, and internal consistency. Questionable values are manually checked and corrected respectively the quality label is adjusted. A final quality check step occurs after all of a month's data are available, focusing on aggregate values. The quality flags are stored in the database and are also made available to users, e.g., when the data are made available on the Internet. DWD quality assurance also includes the identification and correction or description of errors in the historical data Kaspar, F., et al (2013). Appropriate procedures have been developed for the quality control of historical data and are applied to the data. The focus is on the daily values. In general, the quality of these values can be considered reasonably good, but there are still doubtful values on the order of about 0.1-1%, especially for the pre-1979 data. The user must keep in mind that the data can be affected by certain non-climatic effects, such as changes in instrumentation or observation time. With few exceptions, the data are reported "as observed", i.e., no homogenization procedure was applied.





As verification data, rainfall derived radar images and discharge observations from the state of Bavaria were utilized. The radar data used is the product RADOLAN-RW that is provided by the DWD in hourly and daily resolutions starting the year 2005 (DWD Climate Data Center (CDC), 2021). These products have been gauge-adjusted with the observed hourly station data.

The discharge data were provided by the environmental agency of Bavaria with hourly and daily resolutions for approxi-

mately 400 gauges within the region of Bavaria. Different headwater catchments were derived and selected for validating the results. The aim is to identify if a reaction in the discharge was noted after the event or not.

Figure 1 illustrates the location of the sub-daily and daily rain gauges as well as their spatial density. For the daily data all available locations (historical and present) are displayed. This is because all available data were investigated. The spatial density was calculated using a kernel density estimation with a Quartic shape and a radius of influence of 30 kilometers. The

stations do not have a homogeneous spatial distribution over the country. Some locations have a higher network density than others. Figure 2 describes the number of available station over time with the corresponding available data duration.

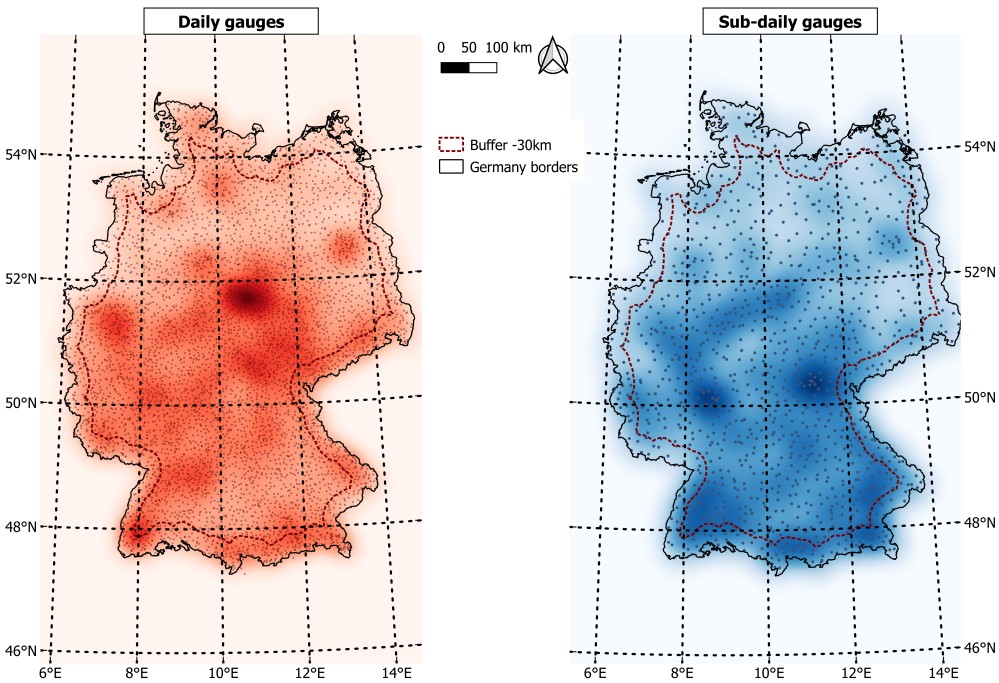

**Figure 1.** Map of the study area showing the location and density of the DWD gauges with daily (left) and sub-daily (right) resolutions.



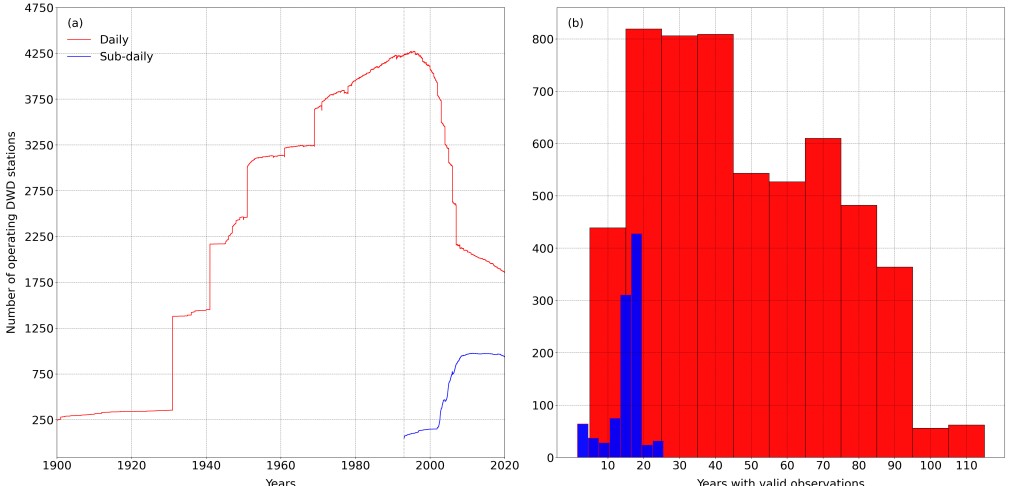

**Figure 2.** Development of DWD precipitation station network. Panel (a) shows the number of available stations with time and panel (b) shows the distribution of the data availability for daily (red) and sub-hourly (blue) data.

## 3 Methodology

### 3.1 Data transformation

As an initial step, a Box-Cox transformation was applied to reduce the effect of the skewed precipitation distribution (Box
and Cox, 1964). To find which transformation factor is most suitable, several simulated truncated standard normal distribution (truncated at $p_0$) were fitted to the original data and the probability of having a value above or below $p_0$ is derived ($p_0$ probability of having 0 mm precipitation value).

$$P(X < a) = p_0 \tag{1}$$

From this probability (denoted $p_{norm}$) a new standard normal distribution is generated where ($x < p_{norm} = 0, x >= p_{norm} = x$). From this distribution the skewness $\gamma_{norm}$ is calculated. The goal now is to find which transformation factor minimizes the
difference between the original data skewness and $\gamma_{norm}$. This was done for each station separately and for all aggregations. Eventually an average transformation factor (denoted hereafter $\lambda$) was derived for each temporal aggregation. The results of this procedure can be seen in figure 4 and Table 2.

Once $\lambda$ was calculated, the original precipitation data was transformed as in equation 2, and in the the newly truncated
normalized space the following approach was implemented to find outliers in the precipitation data over several temporal resolutions.

$$Z^*(u,t) = Z^{\lambda_t}(u,t) \tag{2}$$

Where:





$Z^* =$     transformed precipitation data at location $u$ and temporal aggregation $t$

$Z =$     original precipitation data at location $u$ and temporal aggregation $t$

$\lambda_t =$     transformation factor for temporal aggregation $t$

## 3.2 Outlier detection

The proposed method was initially tested for identifying outliers in groundwater quality data (Bárdossy and Kundzewicz, 1990). In this paper a similar method was implemented to identify unusual precipitation data and is extended by a validation of the results using external information such as radar or discharge observations. For detecting precipitation records that are possible outliers the concept of jackknifing is used. A method initially developed by (Quenouille (1949) and Quenouille (1956)). The main idea is based on removing one (or each) observation from the data and estimating its value again. In this study, for checking the quality of intense precipitation values, the largest four yearly observations for every station are compared to the estimated values at the same location. For example, if a station has 5 years of data, 20 events were investigated (for every aggregation). Many possible faulty observations can only be detected if they are inspected at lower temporal aggregation and on not on the temporal scale at which they were observed. This is the case when looking at sub-daily and sub-hourly values were a single observation might not be unusual but the accumulation of many values reveals suspicious sums.

Ordinary Kriging (OK) is used as an interpolation technique. Each cross-validated value is estimated using the nearest 30 neighboring locations with valid observations. From these, the variogram is derived in the rank space domain and rescaled to the variance of the data. This allows a variogram calculation in a more robust manner (Lebrenz and Bárdossy, 2019). The target location is calculated by solving the kriging equation and the estimation variance is noted. For identifying unusual observations the ratio between the absolute value of the difference between the observed and the estimated values and the estimation variance is calculated. This Criteria Ratio, denoted hereafter as $CR$ describes the relative agreement/disagreement between the observed value and the spatial surroundings for the corresponding time step. Larger $CR$ values reflect high spatial-temporal disagreement and low values denote greater agreement. Based on the $CR$ value, different types of event can be identified, namely those occurring on a local scale with high $CR$ values and other on a regional scale with low $CR$ values. As in Bárdossy and Kundzewicz (1990) a $CR$ value of three is initially used to identify suspicious observations. The $CR$ value is derived for every cross-validated event. Eventually the $CR$ value is related to all of the observed (interpolated) data establishing a possibility to find a suitable $CR$ value for identification of precipitation outliers.

$$CR_i(u) = \frac{|Z_i^*(u) - Z_i(u)|}{\sigma_i(u)} \tag{3}$$

Where:

$Z_i^*(u) =$     estimated value at location $u$ and timestep $i$

$Z_i(u) =$     observed value at location $u$ and timestep $i$

$\sigma_i =$     kriging standard deviation at location $u$ and timestep $i$



Since precipitation events occurring on a local scale might represent a single event and not a false one, to validate the first or the second case, the suspicious events are compared to the observed radar or discharge values in the corresponding catchment.

As a final step, the time series before and after the occurrence of the observed value and its neighbors is investigated. If the observed value is due to a 'single' peak or many 'continuous' small peaks the possibility of being an outlier increases. Some outliers can be easily identified as erronneous values other are more challenging, this is were the additional information is valuable.

Following is the implemented space-time precipitation outlier detection scheme

1. for every station select the largest 4 yearly values

2. for every selected observation

3. transform the target and surrounding locations using equation 2

4. calculate the estimated value at this point using the surrounding stations and their spatial structure

5. for the estimated value find the corresponding estimation variance (or standard deviation)

6. calculate the $CR$ value

7. compare the $CR$ value to the tested observation

8. find all selected observations that have high $CR$ values and are within the upper 1% quantile

9. compare these events to the corresponding radar image or discharge values

10. repeat the procedure for different aggregations

11. find events that are suspicious for single or several aggregations

### 3.3 Interpolation

For interpolating the selected events Ordinary Kriging (OK) is used as an estimation technique. OK is part of Geostatistics which refers to multivariate statistics for neighboring values in space.

The main concept behind Geostatistics is the consideration of the data as spatially dependent random numbers with a variance that increases with increasing separation distance. The observed data at the corresponding locations are seen a realization of the regionalized variable of the random space function. Since for every location $u$ in the domain $D$ there is many (infinite) random variables $Z(u)$ describing each $Z(u)$ using it's own distribution function $F_Z(u)$ is practically impossible. For simplifying the problem different hypotheses are considered. The first hypotheses which is a central one in Geostatistics is stationarity. Simply said, the whole domain $D$ is represented by a single distribution function regardless of the location of the points $u$ in $D$. A further simplification is introduced with the second-order stationarity. For this the expected value of the random function $E(Z(u))$ is constant over the domain and the covariance of two random variables corresponding to two locations





$u_i$, $u_j$ depends only on the separating vector $h = u_i - u_j$ between the two points. This means that the covariance depends
on the spatial configuration of points are and not their exact values. The second-order stationarity hypotheses requires that a
covariance function exists. Since for a separation distance of $h = 0$ the covariance is same as the variance, the existence of
a finite variance for $D$ is required. This is often not the case in many natural processes (such as rainfall) where the variance
increases with the distance. To solve this problem, the final hypothesis known as the intrinsic hypothesis was introduced.

Same as the second-order stationarity the expected value is constant all over the domain D and the increment of the variance
between two locations depends only on the separating vector $h$. The intrinsic hypothesis is a simplification of the second-order
stationarity that is not constrained on the variance but on the variance of the increments.

The (semi-) variogram is defined by:

$$2\gamma(h) = E[Z(u+h) - Z(u)]^2 \tag{4}$$

Where:

$$
\begin{aligned}
Z(u+h) = & \quad \text{observation value at location } u+h \\
Z(u) = & \quad \text{observation value at location } u \\
h = & \quad \text{separation distance}
\end{aligned}
$$

**Properties of the variogram**

For a given separating distance $h$ between pairs of data the variogram needs to have the following properties:

1. $\gamma(h = 0) = 0$ (always 0)

2. $\gamma(h) > 0$ (because it is a square, a variance)

3. $\gamma(h) = \gamma(-h)$ (symmetrical, because it is a squared difference)

4. Variance of the increment is a function of h

5. Asymptotic behavior: there is a kind of a limit of continuity

6. Nugget effect; for very small distance, there are differences

7. Anisotropy: $\gamma(h)$ may differ from one direction to the other

The experimental variogram is derived from the observed values and their spatial distribution. A theoretical variogram model
was then fitted to the experimental one. Once the varioagram was estimated OK could be performed.

OK is a regionalization method to estimate an unknown value at a target location by solving a linear equation system through
minimizing the estimation variance and maximising the accuracy (no systematic error).





The estimation of the target location $Z^*$ using the surrounding observations $Z_i$ at the measurement locations $n$ is defined by a linear estimation equation:

$$Z^* = \sum_{i=1}^{n} \lambda_i Z i \tag{5}$$

The kriging estimation variance at the target location is formulated as:

$$\sigma^2(u) = -\sum_{i=1}^{n}\sum_{j=1}^{n} \lambda_i \lambda_j \gamma(ui - uj) + 2\sum_{i=1}^{n} \lambda_i \gamma(ui - u) \tag{6}$$

The weights $\lambda_i$ are solved by guaranteeing the unbiased property of OK. Namely, the expected value of the estimation value should be equal to the expected value of the field Z. For this the Lagrange multiplier $\mu$ is introduced and the following linear system is solved:

$$\sum_{j=1}^{n} \lambda_j \gamma(u_i - u_j) + \mu = \gamma(ui - u) \qquad \forall i = 1,..,n \tag{7}$$

$$\sum_{i=1}^{n} \lambda_i = 1 \tag{8}$$

Note that $\gamma$ refers to the (semi-) variogram which can also be replaced by the covariance function with the respected modifications. These two functions reflects the change of correlation as a function of the separating distance between the spatially distributed values. Note that for estimating the variogram the cross-validated location is not used.

     For the case where all neighboring stations had a zero precipitation value, an theoretical spherical variogram with a range of 30 km was used.

Figure 3 shows a flowchart describing the applied method starting with the data download procedure and ending with the identification of suspicious observations.

### 3.4   Data corruption

To test further the validity of the method, twenty stations without detected outliers were randomly selected and their data (same events as before) were 'artificially' contaminated. The transformed observations of each target location were decreased and
increased by several percentages (from 5 to 150 %) and the outlier detection method was tested. The main idea here is to check if the method is able to identify such previously non-existent false values. The results of this procedure can be seen in Table 1.





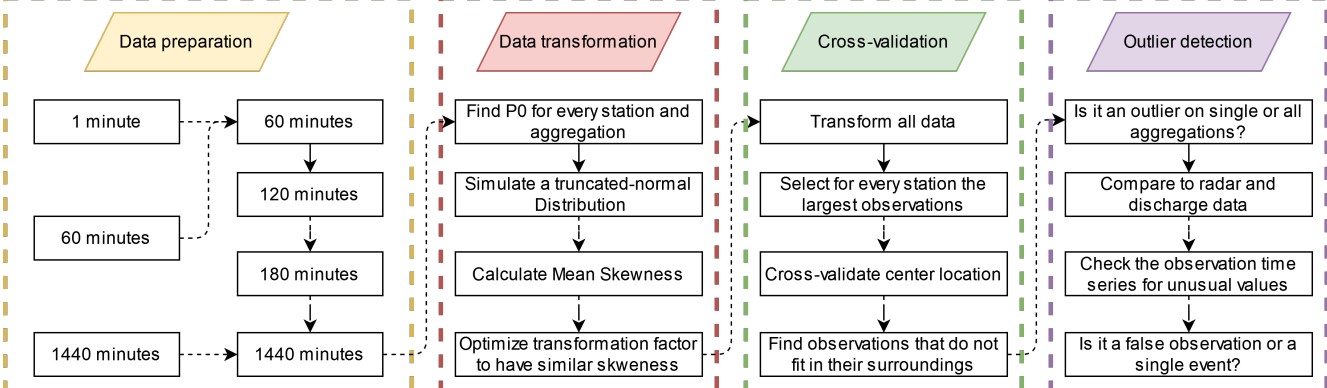

**Figure 3.** Flowchart summarizing the described method.

**Table 1.** Number of newly detected events after corrupting by different percentages the cross-validated observations of 10 randomly selected stations with no previous outliers.

| | Temporal aggregation | 60 min | 120 min | 180 min | 240 min | 360 min | 720 min | 1440 min |
|---|---|---|---|---|---|---|---|---|
| | **Number of events** | **150** | **150** | **150** | **150** | **150** | **150** | **150** |
| | **Minimum of the minima [mm]** | 5.12 | 5.16 | 5.17 | 5.13 | 5.26 | 5.6 | 5.17 |
| | **Average of all averages [mm]** | 11.05 | 12.67 | 14.41 | 14.41 | 16.85 | 19.8 | 24.03 |
| | **Maximum of the maxima [mm]** | 51.2 | 50.1 | 53.47 | 63.93 | 71.92 | 73.6 | 76.37 |
| | **-100 [%] (false zero)** | 88 | 115 | 102 | 125 | 100 | 124 | 141 |
| | **-50 [%]** | 10 | 29 | 38 | 41 | 33 | 46 | 65 |
| | **-25 [%]** | 2 | 3 | 13 | 9 | 8 | 8 | 4 |
| | **-10 [%]** | 0 | 0 | 0 | 0 | 0 | 0 | 1 |
| | **-5 [%]** | 0 | 0 | 0 | 0 | 0 | 0 | 0 |
| Percentage of error | **0 [%]** | **0** | **0** | **0** | **0** | **0** | **0** | **0** |
| | **5 [%]** | 3 | 1 | 2 | 4 | 10 | 3 | 2 |
| | **10 [%]** | 4 | 11 | 6 | 12 | 20 | 7 | 9 |
| | **25 [%]** | 23 | 45 | 48 | 52 | 65 | 46 | 55 |
| | **50 [%]** | 74 | 88 | 118 | 121 | 119 | 113 | 116 |
| | **100 [%]** | 149 | 150 | 150 | 148 | 150 | 149 | 149 |
| | **150 [%]** | 150 | 150 | 150 | 150 | 150 | 150 | 150 |





# 4 Results

## 4.1 Transformation to truncated normal

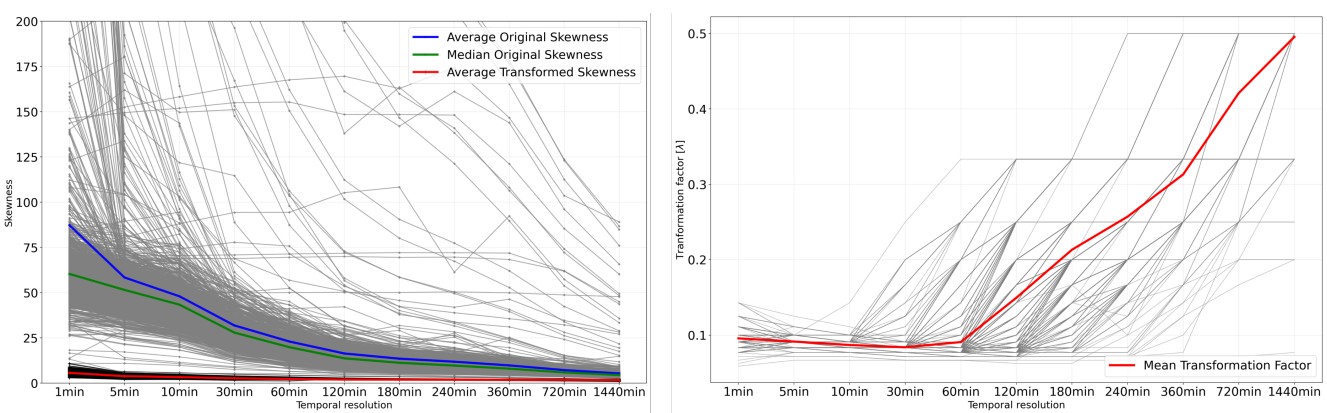

**Figure 4.** Left panel shows the skewness values along the mean and the median of all stations before and after transformation . Right panel shows the average transformation factor $\lambda$ derived from all stations and for each temporal aggregation.

**Table 2.** Average transformation factor $\lambda$ used to transform the original data to the truncated normal space with reduced skewness.

|  | 60 min | 12 0 min | 180 min | 240 min | 360 min | 720 min | 1440 min |
|---|---|---|---|---|---|---|---|
| **Average transformation factor $\lambda$** | 0.097 | 0.155 | 0.219 | 0.262 | 0.318 | 0.427 | 0.499 |

## 4.2 Outliers vs single events

Based on the CR value, different events can be identified. Left panel of figure 5 represents the CR value versus the ratio between the interpolated and observed values. All values denoted in red have a CR value above 3. This figure allows identifying the events that are of interest and relating the CR value to the interpolated and observed data. Note that the observed and interpolated values are in the original non-transformed space, only the CR values are calculated from the interpolation of the transformed values. The suspected outliers (high CR values) are further inspected.

The values in the plot having a ratio of interpolated to observed of 5, are values obtained by interpolating with the original values where a neighboring station had simultaneously recorded an outlier (in this case a false high observation).

The cumulative distribution function (CDF) from all investigated observations was calculated and the location of the detected outlier on this cdf are marked. This can be seen in the right panel of Figure 5. The events that were detected as being outliers spread over the curve showing that the method can detect not only high values but as well relatively small values that differ

highly from their neighboring space.





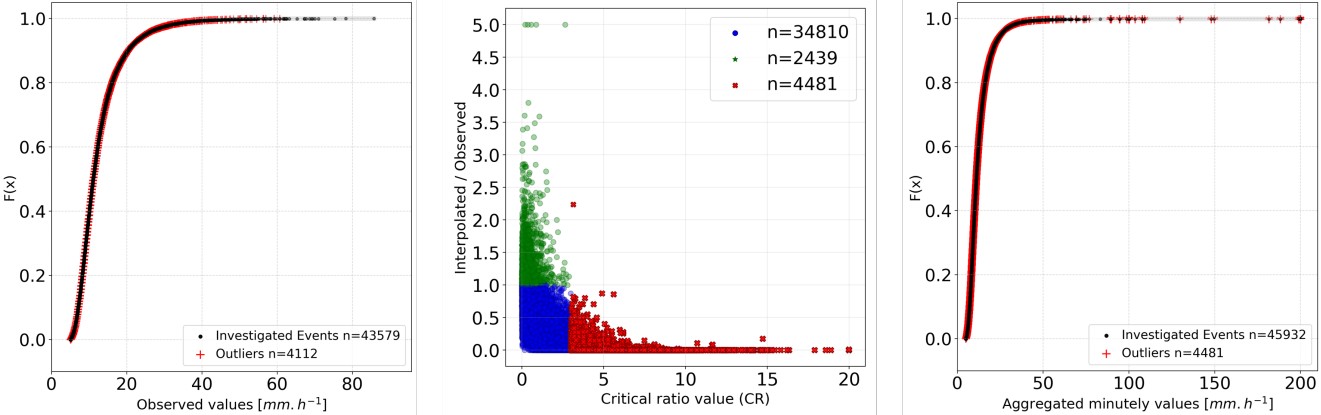

**Figure 5.** The left panel shows the CDF of all investigated hourly events with the detected outliers marked in red. Center panel shows for the minutely aggregated data the CR values versus the ratio of interpolated and observed hourly values in the original data space. The right panel shows the CDF of all investigated hourly events with the detected outliers marked in red. The hourly data in the centre and left panel were aggregated from the minutely values. Note that an upper limit of 200 mm h$^{-1}$ was set.

## 4.3 Selected case studies

In this following part selected events, that were identified as outliers are presented. The first example in figure 6 shows the presence of unusual values in the minutely data of the cross-validated station ($> 8\ mm\ min^{-1}$). The radar data for that hour are used for result verification and do not show such a high-intensity event that occurred above the investigated location. The

second example in figure 7 shows a similar case in the minutely data but the radar image confirms the occurrence of the event.

In small river catchemnt, discharge data can also be used to identify the occurrence or absence of an identified outlier. To this purpose, discharge data from smaller headwater catchments in the federal state of Bavaria with one (or many) rain gauge stations within the catchment were analysed. If a rain gauge observation was identified as being suspicious the discharge values following the event were checked. An example for is, is shown in the upper Pegnitz catchment which is located on the

northern part of the Bavaria (Fig. 8). Panel b) in figure 8 shows an hourly outlier observation that resulted in a reaction in the corresponding headwater catchment. On the other hand, panel c) in figure 8 shows the opposite case, i.e. an hourly outlier that did not cause any reaction in the corresponding catchment. Note that both cases are selected for the same gauging station in the Pegnitz headwater catchment.

## 5 Discussion and conclusion

Through this work a methodology to identify outliers in intense precipitation data was presented. From an hourly to a daily temporal aggregation the largest four yearly values for every station were identified and analysed. The method is based on a relatively simple interpolation technique in a cross-validation mode.





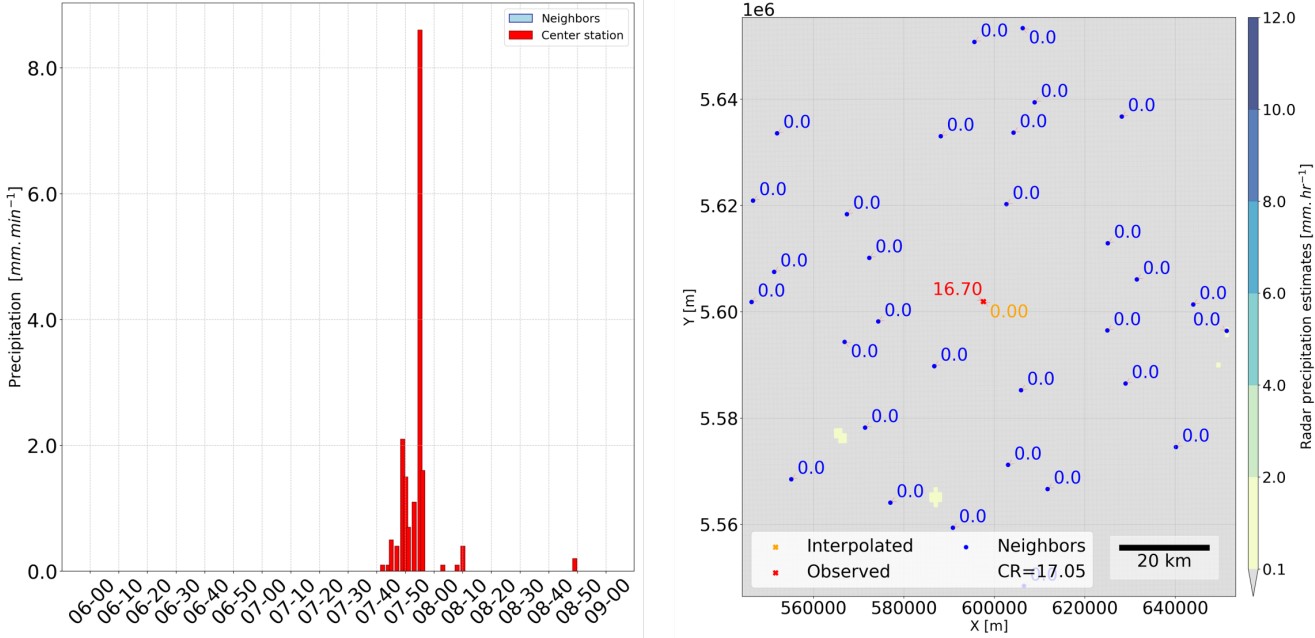

**Figure 6.** Example of an identified outlier in the hourly values that occurred on the 14 May 2008 between 7:00 and 8:00. Left panel shows the minutely observations of the target and neighboring stations. The right panel shows the spatial distribution of the stations and the corresponding radar image from the hourly RADOLAN-RW values.

**Table 3.** The diagonals show the number of unique days with identified outliers. The other values reflect the number of common days between the reference and test aggregation. For example, for the reference aggregation 60 min and the test aggregation 120 min, there are 1223 days with identified outliers that are in the reference and in the test aggregation.

| | | Test aggregation | | | | | | |
|---|---|---|---|---|---|---|---|---|
| | | **60 min** | **120 min** | **180 min** | **240 min** | **360 min** | **720 min** | **1440 min** |
| Reference aggregation | **60 min** | **1581** | 1223 | 1189 | 1167 | 1083 | 819 | 683 |
| | **120 min** | 1223 | **1441** | 1231 | 1204 | 1087 | 795 | 654 |
| | **180 min** | 1189 | 1231 | **1533** | 1293 | 1192 | 876 | 708 |
| | **240 min** | 1167 | 1204 | 1293 | **1604** | 1275 | 947 | 767 |
| | **360 min** | 1083 | 1087 | 1192 | 1275 | **1622** | 1063 | 860 |
| | **720 min** | 819 | 795 | 876 | 947 | 1063 | **1590** | 1151 |
| | **1440 min** | 683 | 654 | 708 | 767 | 860 | 1151 | **1572** |

To cope with the positively skewed rainfall distribution a Box-Cox transformation was applied. The transformation factor was derived by fitting a truncated normal distribution to the original data and by optimizing the factor to have similar skewness values. The factor is averaged from all stations and is calculated for each aggregation. The difference between the median and


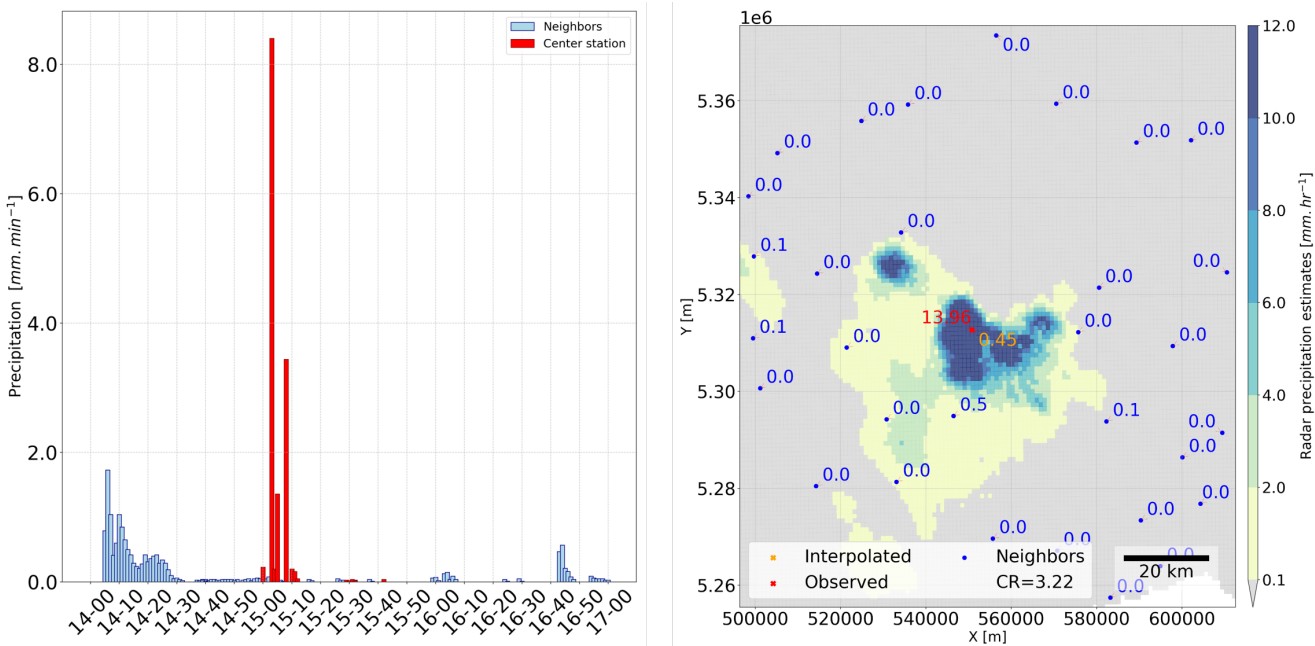

**Figure 7.** Example of an identified outlier in the hourly values that occurred on the 22 of June 2005 between 15:00 and 16:00. Left panel shows the minutely observations of the target and neighboring stations. The right panel shows the spatial distribution of the stations and the corresponding radar image from the hourly RADOLAN-RW values.

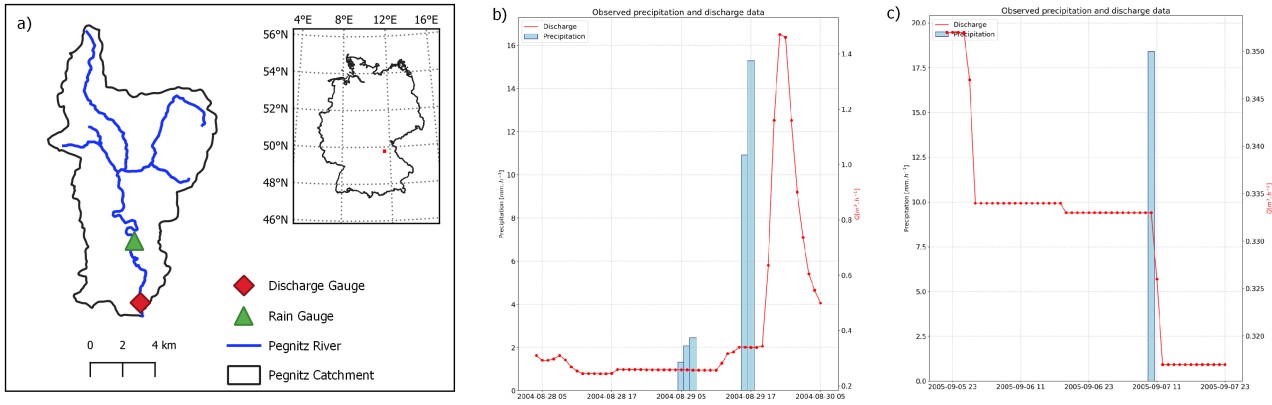

**Figure 8.** a) Location of discharge and rain gauge station within the Pegnitz headwater catchment (panel a) and observed discharge and precipitation data (+/- 1 day) for detected outliers with b) a discharge increment and c) without a discharge increase

the mean skeweness reveals that some stations have very high skewness values affecting the mean on the sub-hourly scale. Once the factor have been derived, the transformation of the data and the subsequent cross-validation were applied.





**Table 4.** The diagonals show the number of unique days with identified outliers. The other values reflect the number of different days between the reference and test aggregation. For example, for the reference aggregation 60 min and test aggregation 120 min, there are 358 days with identified outliers that are in the reference and not in the test aggregation.

| | | Test aggregation | | | | | | |
|---|---|---|---|---|---|---|---|---|
| | | **60 min** | **120 min** | **180 min** | **2 40 min** | **360 min** | **720 min** | **1440 min** |
| Reference aggregation | **60 min** | **1581** | 358 | 392 | 414 | 498 | 762 | 898 |
| | **120 min** | 218 | **1441** | 210 | 237 | 354 | 646 | 787 |
| | **180 min** | 344 | 302 | **1533** | 240 | 341 | 657 | 825 |
| | **240 min** | 437 | 400 | 311 | **1604** | 329 | 657 | 837 |
| | **360 min** | 539 | 535 | 430 | 347 | **1622** | 559 | 762 |
| | **720 min** | 771 | 795 | 714 | 643 | 527 | **1590** | 439 |
| | **1440 min** | 889 | 918 | 864 | 805 | 712 | 421 | **1572** |

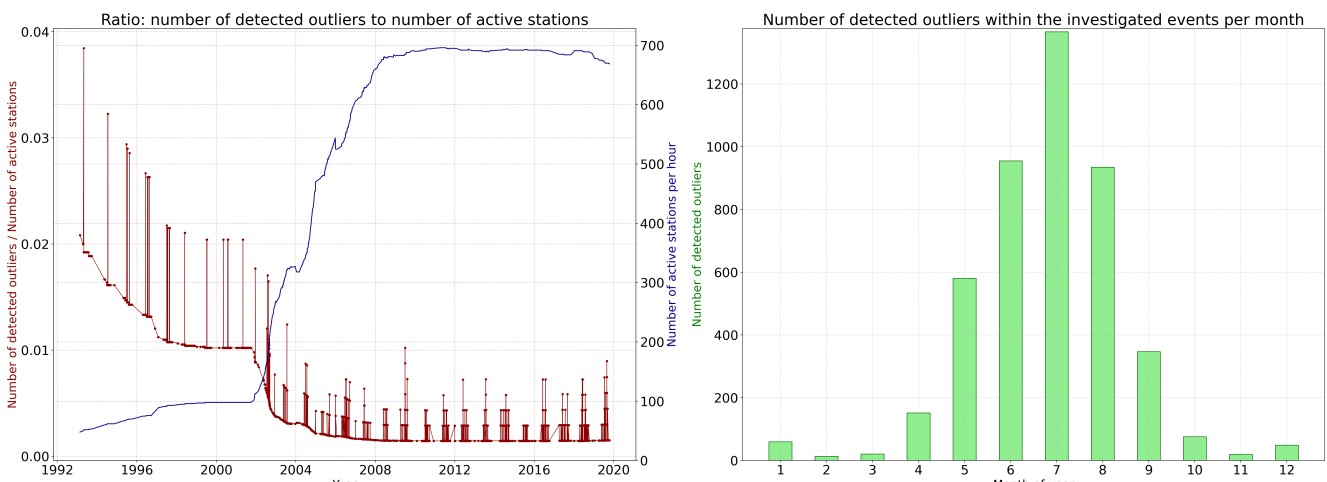

**Figure 9.** Left figure shows the number of hours with outliers within the investigated hourly events (aggregated from 1 minute) of all stations pro year. The right figure shows the number of detected outliers within the investigated events for every month within the hourly data using a CR value of 3.

The results revealed different events throughout various temporal aggregations that strongly differed from their surroundings at the same observation time. Some events were identified as being outliers over several temporal aggregations, while other events appeared/disappeared with change of temporal aggregation. Since several datasets were present, namely the minutely, hourly and daily data, a cross-investigation was done to test if unusual outliers in the different datasets are similar.

For events were neighboring stations had false extreme observations, the interpolated value in the original data space (without transformation) is influenced by these values and often exceeds by much the observed value. Depending on how many


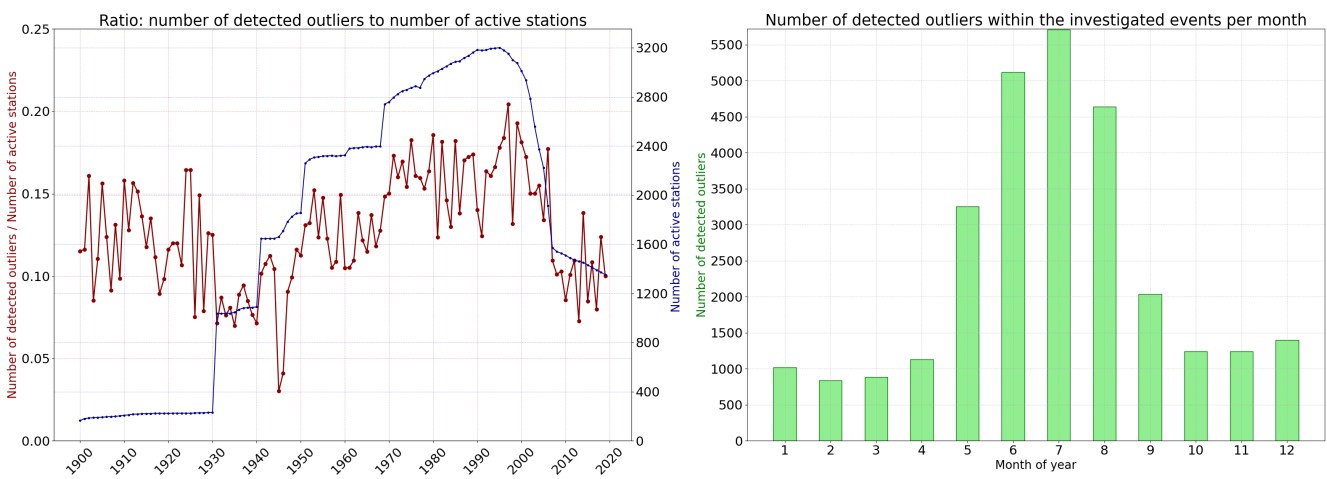

**Figure 10.** Left figure shows the number of days with outliers within the investigated daily events of all stations pro year. The right figure shows the number of detected outliers within the investigated for every month within the daily data using a CR value of 3.

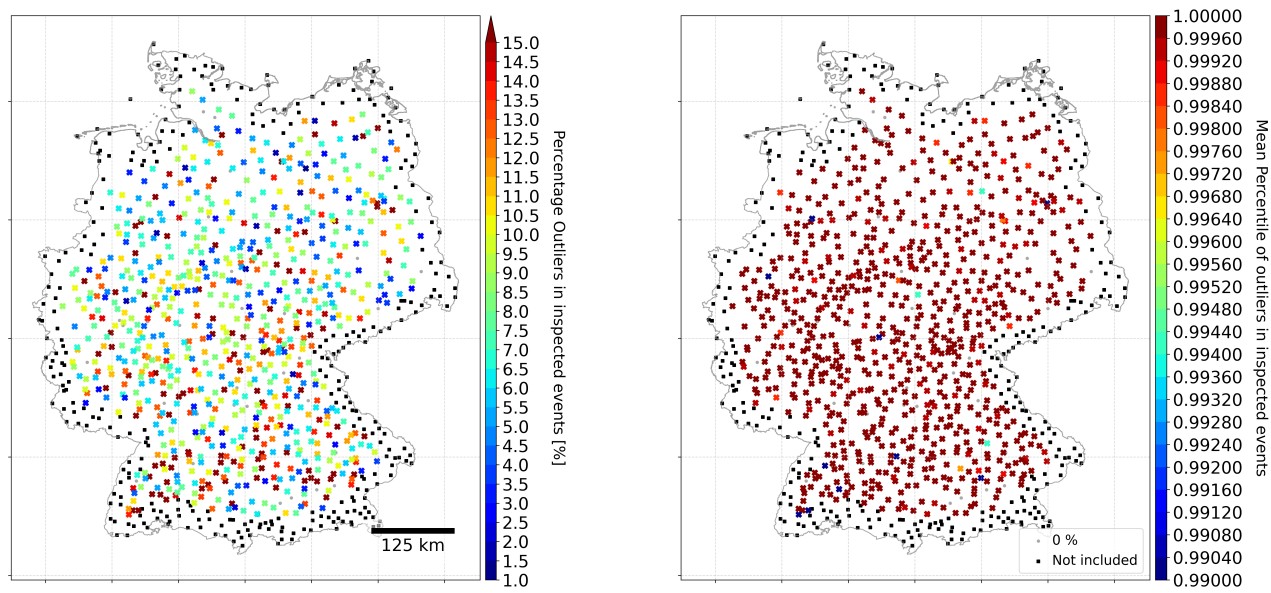

**Figure 11.** The left figure shows the percentage of possible outliers and the right figure shows the corresponding mean percentile in the hourly station data for a CR value of 3. The locations denoted in grey are stations with no found outliers for this CR value.

neighboring observations have simultaneously false observations (which is rarely the case) the kriging standard deviation in
the transformed space is relatively low.





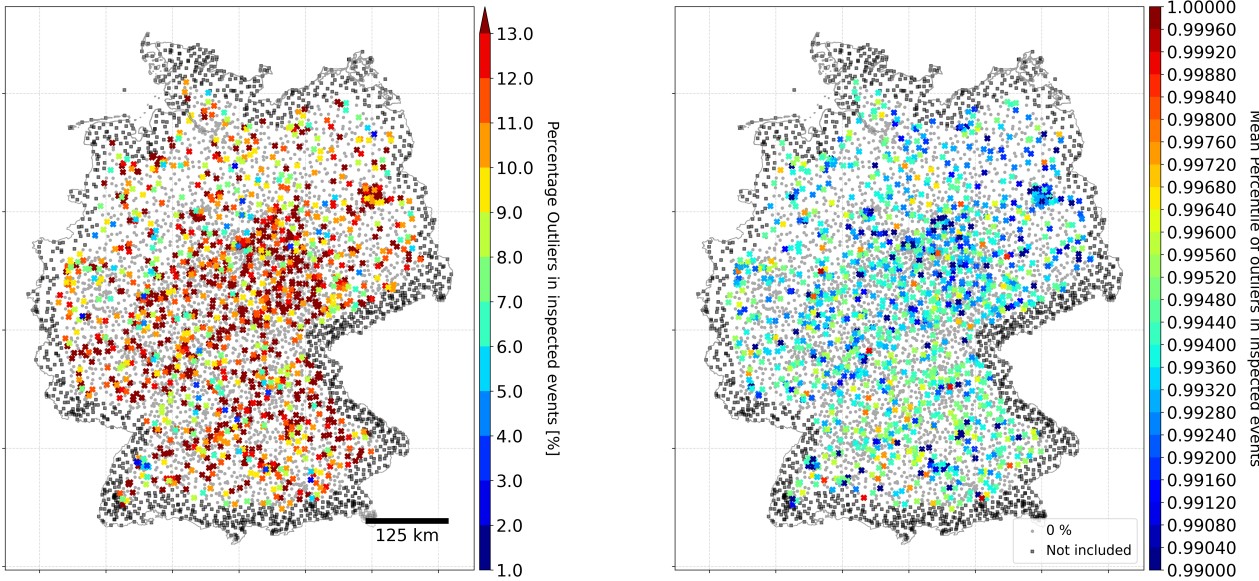

**Figure 12.** The left figure shows the percentage of possible outliers and their corresponding mean percentile (right figure) in the daily station data over the region of Germany using a CR value of 3. The locations denoted in grey are stations with no found outliers for this CR value.

For several stations no outliers were detected in their data. To test the robustness of the method, their data was corrupted with several percentages and checked again. The method was able to identify most events as outliers as the value of the added error increased. By decreasing the observed value until reaching a false zero observation the method was able to identify around 60% of the cases as being outliers on the hourly scale. With increasing temporal aggregation the identification of false zeros increased reaching 94% on the daily scale. By increasing the error value to up to 100%, almost all values were detected as being an outlier on all temporal aggregations.

Note that in this case, different events on each timescale were investigate but there is the possibility to investigate each event on all temporal aggregations.

When dividing the data based on the month in which they occurred, the number of identified outliers in the summer period is much larger than in the winter one. This was noted on all temporal aggregations and was presented for the hourly and daily aggregations. This is related to convectional rainfall procedures occurring especially in the period between June and August where several events occur on a local scale. Such events are detected as outliers and are not necessarily false observations.

The detected events are denoted as being suspicious as they can either be a false observation or a single event. For distinguishing between the two possibilities additional external data has been used. Discharge gauge data of corresponding headwater catchments and radar rainfall images were used when available. A change in the water level (or discharge value) within a time interval after the event date revealed the event as being a single event and not a false value. The radar images for the corresponding time step (or for the accumulated sum) were used to find the presence or absence of a rainfall event over the station





location. A final choice regarding flagging an observations is done carefully and individually for every location. Eventually the flagged observations are kept aside and investigated before being used in the further analysis.

The current method needs to be extended and modified for temporal aggregations below the hourly scale. Especially the kriging methodology should include time as a third dimension to account for advection and correlation between subsequent steps. Moreover, many events are identified as being an outlier when part of the neighboring stations had zero precipitation values. This can happen in case of directional dependent events driven by a frontal system. This cases could be further handled by including anisotropy in the interpolation method.

The aim of this study was to develop a relative simple method to check the intense observed rainfall values and identify unusual observations that should be carefully handled.

*Data availability.* The precipitation data was obtained from the Climate Data Center of the Deutscher Wetterdienst (https://opendata.dwd.de/climate_environment/CDC). The discharge data are provided by the environmental state agency of Bavaria LfU (https://www.gkd.bayern.de).

*Code and data availability.* The corresponding code is available upon request from the contact author

*Author contributions.* AEH developed and implemented the algorithm for the study area. AB designed and supervised the study. All authors contributed to the writing, reviewing and editing of the manuscript.

*Competing interests.* The authors declare that they have no conflict of interest.

*Acknowledgements.* This study is part of the project B 2.7 "STEEP - Space-time statistics of extreme precipitation" (Grant No. 01LP1902P) of the "ClimXtreme" project funded by the German Ministry of Education and Research (Bundesministerium für Bildung und Forschung, 
BMBF). The authors thank the German Weather Service DWD for providing the precipitation and weather radar data and the Bavarian Environment Agency for providing the discharge data. Moreover we acknowledge all developers of different python core libraries (e.g. numpy, pandas, matplotlib, cython, scipy) for providing open source code. The authors thank the University of Stuttgart for funding this open-access publication.





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
