# Peer review of "Technical Note: Space-Time Statistical Quality Control of Extreme Precipitation Observations"

_Hydrology and Earth System Sciences, 2022_

## Referee Comment (RC2)

[referee-annotated manuscript omitted]

---

## Author Response (AR1)

Response to Reviewer #1

We thank the first reviewer for taking his time to review our manuscript and providing important remarks. New comments are added in blue.

General Remarks

The authors give a good explanation of the statistical methodology, but there are some

shortcomings in the applied section:

- the ancillary data are introduced "as is" without acknowledging their quality

Ancillary data is referring to the radar and discharge data. Regarding the discharge data, our approach is to check whether a precipitation event has led to a reaction in runoff or not. Therefore, the accuracy of the runoff values is of minor importance. Hence, we take the discharge data of the provider(s) without performing any further plausibility checks. The case with the radar data is similar, we are aware that precipitation estimates form radars are prone to errors but the case here is similar do the one with the discharge data. We primarily use this data to check whether precipitation has occurred at a specific site or not.

*Since our goal was to use the ancillary data to verify if a detected outlier was a false observation or a single event, the exact quality of the ancillary data does not play a major role. The data were already quality checked by the DWD (Radolan-RW) and the by the LfU (discharge data)*

- there are several figures which are not cited but part of the contribution: here clearly

textual explanation is missing I assume that the authors have a clear idea about the value of the figures, and they should share these ideas more explicitly with the readers

Thank you for your remark, indeed more explanation is needed. Some figures or tables are described shortly in the discussion section without proper referencing. This will be improved in the revised manuscript.

*We reduced the manuscript size to meet the requirements of a technical note, therefore we reduced the number of figures and tables. Moreover some figures were combined together. In the new manuscript all presented figures and tables are mostly discussed in the results section.*

For this reason, a careful major revision should be performed in order to increase the quality of the description for the practical section up to the quality level of the theoretical section.

Specific Remarks

line 93: RW data have two properties which need to be discussed before use:

- they are hourly data

- the quality is varying over the years

*As mentioned before, the RW data are used as qualitative information if a strong precipitation event has occurred, therefore the QPE and the change of quality with time play a minor role.*

line 99: the term "kernel density estimation with a Quartic shape" deserves more explanation for a better understanding

*This was done in line 90 of the manuscript.*

Figure 1: an explanation for the colouring with a scale would be useful

*This was added to figure 1.*

line 147: the quantitative criteria for comparison with radar data and with discharge data should somewhere be explained

When comparing with radar data, we considered that the occurrence (or absence) or precipitation observation over the target location is an indication to the quality of the observation. As for the discharge data, we expect a reaction in the headwater catchment discharge (within few hours) after the event occurred in case of correct observations. This will be added to the manuscript in the methodology section.

*This was added in line 80 and 85 of the new manuscript.*

line 152ff: a flow chart would be more helpful for the reader to follow the steps of the algorithm. In particular, the limit of loops is not obvious in the current list.

*The step-by-step procedure was removed and the flowchart was modified and added instead in figure 2.*

Section 4.1: a discussion of figure 4 and table 2 is missing here - or move the figures back to section 3.1

*As suggested we moved the table to section 3.1 and added the figure was removed*

line 238: why are minute data here directly comparable to hourly sums? What are your criteria and where do you expect them to have limitations?

*The aim is to check if the quality controlled hourly data have better observations than the minutely data. These are more susceptible to false observations. Spatial consistency is checked more intensively by DWD for higher aggregated precipitation data ( >= 1 h) than for high temporal resolution data (e.g. 1 min). The results showed this as well (figure 3). The aim is to check if we have same outliers for the same station with focus on the high values.*

Tables 3 and 4 have never been cited or explained! Please add appropriate sections for explanation and discussion!

*Table 3 and 4 are now combined (table 3) and a suitable explanation is provided in the results section.*

Figures 9 to 12 have never been cited or explained! Please add appropriate sections for explanation and discussion!

*We combined figure 9 and 11 in figure 6 and added the corresponding explanation, figures 10 and 12 are not shown anymore*

lines 266+267: please be more precise in your description.

*The statement was moved to line 150 and is supported by the results in table 2.*

line 275: the statement should be supported by at least a figure

*Center panel of figure 7 supports this statement, which has been added to line 195*

line 280-281: what length of time interval? Please expand on this!

*Since the headwater catchments have small areas and thus quick response times, we expect a reaction in the discharges within the coming hours after the precipitation event was observed. This was added to the manuscript in line 180.*

Detailed corrections

*All other corrections were added to the revised manuscripts.*

Response to Reviewer #2

We thank the reviewer for taking the time to review our manuscript. Our reply to

the reviewer's comments are as follows: *new comment are added in blue*.

Overall this could be a very useful contribution to HESS if the authors would indicate how the proposed procedure may generalize to other observing systems (e.g synop temperature, wind and humidity observations).

Our method is specifically designed for data with skewed distribution and a high spatio-temporal variability. Outlier checks for other parameters such a temperature are much more straight forward to implement. Since the title specifically mentions precipitation we don't see the necessity to discuss how this could be used for other parameters.

Furthermore the text is written almost completely from the hydrolocical engineering point of view: the authors should remember the E in the journals title, namely that other Earth System scientists should also be addressed and not to be scared off.

See reply above, we explicitly present a quality check for precipitation extremes.

Finally the text is largely written in a funding - interim report style eg about one third of the figures is just added to the text without any discussion or even referencing in the text. All these points (and those commented in the annotated pdf document) should be revised before publication in HESS can be considered.

We will provide and discuss all the figures in the manuscript, this was also pointed out by referee

1. Regarding the "interim report style" we submitted this as a technical note and not a full-fledged scientific paper, therefore we consider a shorter, more report like text to be justified. Regarding the comments in the manuscript (supplement), we will implement/correct/address them accordingly.

There are however some remarks which we would like to comment:

Line 21 ff. This remark is correct, but since the DWD gauges assumed to be installed according to the WMO guidelines, we can rule out such scenarios to a large degree.
Line 74 ff. Of course, there are other weighing gauges, but we specifically refer to those used by the DWD.
Line 153: A flow chart is already presented in Figure 3.
Line 195: We fitted a spherical or an exponential variogram model with no nugget to the experimental one. Both models guarantee that the kriging covariance matrix is positive definite. Moreover we will add a section showing that the choice of the variogram model has little influence on the results

Response to comments in the pdf:

line 1: we agree with the comments but we consider hydrological extremes and especially precipitation extremes as rare events since they occur rarely

line 3: quality check was replaced by quality control

Line 4: our goal is to check precipitation extremes, namely the biggest values

Line 16 and 17: we fixed the references.

Line 21: the provided suggestion was added to the manuscript (line 19)

Line 24: change sentence

Line 41: thank you for the suggestion, this was formulated in the manuscript

Line 60: this was corrected.

Line 74-75: the specified gauges are those used by the DWD during the analysis

Figure 1: color bar scale was added

Line 106: this was corrected (see line 104)

Line 112: this was added (see line 96)

Line 131: this can indeed happen and was noted in the results (green point in center panel see figure 3 with a ratio of 5). Line 162 discusses this issue.

Line 151: this was added in the flowchart (figure 2)

Line 169: the remark was correct but we removed the part related to the theory of ordinary kriging to reduce the manuscript length. Corresponding references for ordinary kriging were provided

Line 194: remark also correct but this part was removed and a corresponding reference for the properties of the variogram was given

Line 196: we used a spherical or an exponential variogram with no nugget, both guarantee that the kriging covariance matrix is positive definite and fulfills the properties of the variogram. We tested different variogram models and the results varied slightly with no major implications. This was not added to the manuscript since it is not highly relevant.

Line 221: description of table 1 is given in line 150

The References were corrected.

---

## Author Response (AR2)

We thank the editor and both reviewers for taking the time to review again our manuscript. All suggested corrections were added to the new manuscript.

**Editor's minor comment**

Line 55. Replace the dot in the separation of thousands with a comma (as well as in other places in the manuscript, if needed).

*Line 55. Replace the dot in the separation of thousands with a comma (as well as in other places in the manuscript, if needed).*

*Done*

*Figure 4. Please increase the font size in the figures. Please convert the coordinates to lat/lon as in Figure 1. In Figure 1, consider adding a small box to refer to the area zoomed in Figure 4.*

*Done*

*Figure 6. Keeping only panel c in the main text and moving panels a and b into the supplementary material would be a good idea.*

*Since the text already includes a description of panels a and b and are relevant to the results, We found that adding panels a and b to the appendix is more suitable.*

*Code and data availability. It is strongly recommended that the code be made available for everyone to access online. You should facilitate public access to the code since you are publishing an open-access Technical Note. Although this is not a mandatory procedure, it is recommended.*

*We provided opensource code and data in the Github repository qcpcp https://github.com/AbbasElHachem/qcpcp*

**Referee 1 Comments**

one technical correction is necessary: in Fig. 1 although number of stations per 30 km is a density: it is a line density (!!) but apparently an area density is meant giving the units num of stations per $(30\ km)^2$ or number of stations per $(\pi * 30km), needs to be corrected$

*Thank you for this remark, this was corrected in Fig.1*

**Referee 2 Comments**

line 153: is there a reason why you use 30 neighbouring stations? Depending on the network density, there may be substantial differences in local climate or in precipitation in the far distance for this selection criterion.

*30 is the minimum number of points for the estimation of the variogram - not for Kriging. The number 30 is in fact somewhat too big, but it has not much importance as due to the shading effect in Kriging the far-away stations have a very small weight. The previous statement was added partly to the text in line 118.*

lines 263-264: it is not clear to me if you refer to 200 mm within one hour (which is clearly not possible in this climate) or the intensity of 200 mm/h within a short time period such as 5 minutes which may occur. Maybe that you should clearly state if the minutely data are aggregated to hourly data here or to shorter time intervals.

*This was corrected in the text. The values refer to the total accumulated rainfall sum and not the intensity. This was added in line 174.*

*All Detailed corrections were implemented*